# A Taxonomy of Product–Service System Perturbations through a Systematic Literature Review

Hanfei Wang [1,*], Yuya Mitake [2], Yusuke Tsutsui [3], Salman Alfarisi [1] and Yoshiki Shimomura [1]

1   Faculty of Systems Design, Tokyo Metropolitan University, Tokyo 191-0065, Japan
2   Human Augmentation Research Center, National Institute of Advanced Industrial Science and Technology, Chiba 277-0882, Japan
3   Faculty of Computer Science and Systems Engineering, Okayama Prefectural University, Okayama 719-1197, Japan
*   Correspondence: wang-hanfei@ed.tmu.ac.jp; Tel.: +81-070-4340-5108

**Abstract:** Perturbations have a negative influence on the operation of the business system, which may weaken business performance. However, in the field of the product–service system (PSS), perturbation is still a rarely discussed concept. Researchers and managers have a limited understanding of perturbations in the context of PSS. This hinders PSS designers and managers to prepare for mitigation due to a lack of knowledge and information. Thus, this paper aims to build a taxonomy of PSS perturbation through a systematic literature review. To achieve this target, the authors have reviewed 171 papers and found 18 effective papers. Twenty-five items are considered effective ones that are directly related to PSS perturbation. The result of the review shows that PSS perturbations could be classified into six categories, namely, behavioral, social, environmental, competence, resource, and organizational perturbations. The proposed terminology and taxonomy appear to be effective, which could enable researchers to understand the scope of PSS perturbations on a conceptual level. This finding is also expected to provide useful knowledge and information for researchers who are interested in vulnerability analysis and the robust design of PSS.

**Keywords:** product service system; perturbation; taxonomy; systematic literature review; business performance

## 1. Introduction

In recent decades, the manufacturing industry in developed countries has faced the challenges of higher costs and competitive markets. It appears that traditional manufacturing companies cannot keep the business mode they used to own. Thus, servitization is considered a direction to mitigate this predicament, which could improve value and competitiveness by providing additional services (Martinez et al. 2010; Kryvinska et al. 2014). In this context, the product–service system (PSS) has attracted tremendous attention since it could sustainably integrate products and services. According to Mont (2002), PSS is a marketable set of products and services capable of jointly fulfilling a user's needs. Furthermore, it was also pointed out that this system could benefit itself by sharing the ownership of products with customers (Tukker 2004), which has shown great potential in reducing environmental impact and improving competitiveness (Haase et al. 2017).

Despite the promising potential, the problems caused by PSS perturbations are worrying. In PSS, a perturbation is any endogenous or exogenous event that modifies the stated PSS operational conditions (Estrada and Romero 2016). The existence of perturbations could lead to an unwanted change in the PSS. Compared with PSS failure, which focuses on the undesired function of a single actor and item (Kimita et al. 2018), the concept of perturbation focuses on all kinds of events that could lead to the unwanted performance of all components of the PSS, namely service actors, products, tasks, and the whole system. Perturbation does not focus solely on the issue of failure and disruption; it is the integration

of the deterioration of performance, the disruption of tasks, and the collapse of the system (Wang et al. 2022).

In the field of robust product design, perturbation is a similar concept to the noise factor. A noise factor is an uncontrollable and sensitive event that can seriously affect the performance of artifacts (Taguchi 1986). Thus, considering this feature, noise could be considered a type of perturbation with high sensitivity and uncontrollability. The most common categories of noise factors are internal, external, and unit-to-unit noise factors (Taguchi 1986). Given that noise factors are expensive and difficult to control, researchers usually recommend reducing the sensitivity of products toward noise instead of mitigating its influence (Park 1996; Arvidsson and Gremyr 2008). However, for a real business, it has been found that companies usually do not prepare a database about this type of event. Instead, they just collect the loss of events (Creveling et al. 2002). It would be a complex task for researchers to propose a robust design if they cannot understand the cause, effect, and features of a perturbation. A clear understanding of perturbation and noise-related manufacturing is considered a promising solution. For the service industry, the major concern related to perturbation is the disruption of process and deviation in service quality brought by accidental events. The huge loss is caused due to the wrong response of staff and the poor design of the process. Most customers have been found to reject purchasing a service again and show lower satisfaction if they experience a service failure and service recovery (Smith and Bolton 1998). Thus, how to enable the service process or service actors to respond to an accidental event is regarded as a critical mission toward robust service (Weiss and Goldberg 2019).

For perturbation in PSS, several pieces of evidence show that PSS is not a robust system that cannot fight against the influence of perturbations. Multiple PSS providers suffered a huge loss during unexpected accidents, which could be called perturbations. For example, COVID-19 is a typical unforeseen event that has caused considerable economic and social loss to PSS providers. In New York, the shared bike system experienced a sharp reduction in average trips (Padmanabhan et al. 2021). It was also reported that the number of bookings and occupancy rates for Airbnb was significantly reduced in the US during COVID-19 compared to previous years (Boros et al. 2020). The income of Airbnb also experienced a sharp reduction during COVID-19 (Chen et al. 2020). Furthermore, given that PSS is a direction for manufacturing firms who are struggling to survive in the competitive market, some PSS firms are not just organized but also transformed from manufacturing enterprises. Therefore, the unwanted change in performance could happen in a pretty early period of PSS operation due to the inability of the PSS firm. According to de Jesus Pacheco et al. (2019a) and Michalik et al. (2019), many small- and medium-sized manufacturing firms struggle to transform themselves into PSS providers due to the lack of financial resources and management experience. Furthermore, the findings in multiple kinds of literature show that the features of PSS might form unique PSS perturbations. For traditional manufacturers, the theory of 'service paradox' proposes that manufacturing firms may have no experience to manage the risk of becoming a service provider. There is a potential resistance and misunderstanding toward PSS novelty from internal staff, which leads to inadequate efficiency and poor service quality (Brax 2005). The special orientation about ownership is also a troublesome issue. Inagaki et al. (2022) propose that it is highly possible for an early barrier to be related to the features of PSS, especially the special orientation about sharing the ownership of the products. They propose that this would lead to a lack of a sense of responsibility and cultural support, which then cause financial loss and mismanagement. For user-oriented and result-oriented PSS, providers have exposed the inability to control the adverse behavior of customers when there is poor legal support or moral guidance. For example, in the PSS of the shared bike, the destructive behavior of customers is always regarded as an annoying perturbation by PSS providers, leading to broken bikes and low customer satisfaction. PSS providers are at a loss about how to maintain these bicycles and guide customers to use them carefully (Jia et al. 2018).

Based on the above information, PSS researchers and designers were exposed as having poor preparation for perturbation mitigation due to a lack of knowledge related to perturbation identification and management. The substantial loss was caused before they proposed appropriate mitigation. For PSS design, it is important to make a checklist or database about risky events that are related to perturbation. Low completeness of related tasks leads to low performance in PSS maturity (Muto et al. 2015). A knowledge-based design is proposed as a solution to improve the quality of PSS (Akasaka et al. 2012). There is a need to utilize knowledge of perturbation to achieve a robust PSS design. Frustratingly, the useful information that authors could find in the field of PSS is often partial or not enough. Researchers seem to be reluctant to use the term 'perturbation'; this phenomenon was already shown when researchers were reviewing studies related to the vulnerability of PSS (Wang et al. 2022). Instead, the terms 'barrier' (Besch 2005; Kuo et al. 2010; Moro et al. 2020; Inagaki et al. 2022), 'service paradox' (Oliva and Kallenberg 2003; Brax 2005; Dmitrijeva et al. 2022), and 'operational risk' (Reim et al. 2016) have become popular recently. For the description of events that have a risk of bringing unwanted change, there is still no paper integrating the findings of the various aspects of other papers. It is still unclear whether all the events involved in the above concepts would also lead to unwanted changes in the operating performance of a PSS. Furthermore, so far, current research in the field of PSS has not provided a clear explanation of the scope of PSS perturbations. Despite the definition provided by Estrada and Romero (2016), a single definition cannot enable designers and managers to understand perturbation in a complex business environment. Designers and managers have been proven to require further knowledge to overcome challenges during the operation stage (Sjödin et al. 2017). In the real business world, perturbation can originate from various aspects, which require further details about the categories of the various perturbations (Wang et al. 2022). Thus, a detailed taxonomy of PSS perturbation is the critical theoretical basis for understanding the vulnerability of PSS and achieving robustness in PSS. For the above reasons, there is a strong requirement to provide effective and comprehensive knowledge about PSS perturbation through a taxonomy.

To eliminate the misunderstanding and complexity related to PSS perturbation, this paper aims to build a taxonomy of PSS perturbation through a systematic literature review. Given that there is a lack of effective papers related to PSS perturbation in the current PSS field, which might limit the finding of the literature review, this paper uses two further keywords, namely, operational risk and barrier. By identifying useful information through reviewing papers related to the other two keywords, the present paper intends to fill in the research gap, in that none of the literature provides effective knowledge about PSS perturbation features or their taxonomy. This paper aims to provide knowledge of PSS perturbation by integrating knowledge related to different concepts. This paper is expected to provide theoretical support for researchers who want to research the vulnerability and robustness of PSS. Furthermore, this taxonomy could enable researchers who want to build a database for vulnerability mitigation in the field of PSS.

The paper is structured as follows: Section 2 is going to introduce the research background. Section 3 illustrates the methodology to build the systematic literature review and taxonomy. Section 4 explains the categories of the taxonomy. Section 5 discusses the theoretical significance and practical usefulness of this paper. Then, future research direction is also discussed. Section 6 gives a conclusion about the findings and limitations of this research.

## 2. Research Background

### 2.1. PSS and PSS Design

PSS was firstly defined by Goedkoop et al. (1999); PSS is a combination of services and products that form a marketable set of products and services, jointly capable of fulfilling a client's requirements. In 2004, Tukker proposed an important classification of PSS. In this classification, PSS is divided into three types, namely, product-oriented, user-oriented, and result-oriented PSSs. The definitions of the above three types are shown below. A

product-oriented PSS focuses on the services related to the use stage of products. In this type of PSS, services are usually maintenance, consultancy, and advice. A user-oriented PSS focuses on the ownership of products, namely, providing product pooling, renting and sharing, and product leasing. In this mode, providers own products and share them with customers. The major responsibility of user-oriented PSS providers is ensuring the maintenance and supply related to products. A result-oriented PSS focuses on selling the results of machine production by outsourcing or renting this machine. Furthermore, for realizing sustainability, PSS was also given a high level of expectation. It was proposed that a well-designed PSS could improve the eco-environment, save material, and enlarge the life cycle of the involved products (Tukker 2015; Vezzoli et al. 2015).

To achieve the realization of PSS, there are multiple existing methodologies and guidelines that enable designers to design product–service systems (Arai and Shimomura 2004; Morelli 2006; Maussang et al. 2009; Akasaka et al. 2012; Muto et al. 2015; Vezzoli et al. 2017; Sassanelli et al. 2019; Kimita et al. 2022). They provide precious knowledge about how to integrate products and services, indicating many essential design elements. However, it was found that most of the existing methodologies, especially those that originated before 2012, have a poor relationship with industrial practice (Clayton et al. 2012). Furthermore, the perspective of the life cycle was usually overlooked by them (Cavalieri and Pezzotta 2012). Indeed, the actual PSS design is far more complex rather than the design of products and services. PSS design requires holistic consideration of the complex stakeholder network and basic infrastructure (Mont 2002, 2004; Vezzoli et al. 2017). The knowledgebase support for PSS was also seldom mentioned in this period. To improve the above issues, design methodologies after 2012 (Akasaka et al. 2012; Muto et al. 2015; Vezzoli et al. 2017; Sassanelli et al. 2019; Kimita et al. 2022) had a better focus on the problem of practical industry, sustainability, stakeholder networks, and whole-life cycles. Muto et al. (2015) and Akasaka et al. (2012) proposed solution generation tools based on the knowledgebase. Vezzoli et al. (2017) integrated the method of life cycle analysis (LCA) with PSS design, which could fill in the requirement of multiple stakeholders and achieve a sustainable PSS. Kimita et al. (2022) proposed a maturity model for evaluating the maturity level of a firm that desires to develop a service business. Sassanelli et al. (2019) created a methodology called GuRu based on the design for X (DFX) approach, which is useful for evaluating the life cycle and driven by customer requirements. However, these papers also did not validate their effectiveness in enough real cases. Most of the methodologies have only been tested in a single company or single type of PSS mode (product-oriented/user-oriented/result-oriented PSSs). To propose a general design methodology, tremendous practice is required.

Furthermore, given the complexity of PSS, the PSS design process requires further efforts to achieve feasibility, especially in the aspects of innovation, technology, organizational structure, information management, and knowledge management (Sassanelli et al. 2015; de Jesus Pacheco et al. 2019b). Some interesting studies developed PSS designs to become more feasible in real business. To improve the innovation of PSS, Song et al. (2015) built an innovation management framework to manage the innovative issue of PSS. As a famous innovative design tool, TRIZ was also given high-level importance to generate and evaluate innovative solutions for PSS design (Kim and Park 2012; Kim and Yoon 2012; de Jesus Pacheco et al. 2019b). For knowledge management, Baxter et al. (2009) proposed a knowledge management framework to support PSS design based on three key knowledge elements, namely, design knowledge, manufacturing capability knowledge, and service knowledge. This framework showed great effectiveness in the aspects of the manufacturing method and design process. For information management, the condition of products is a major problem for user-oriented PSS. There is a strong requirement for providers to know the actual situation when products are rented by customers (Qu et al. 2016). To solve this issue, Löfstrand et al. (2012) suggested solving this problem from a technical perspective, which utilized a sensor data stream monitoring system to predict the condition of hardware. Teixeira et al. (2012) proposed a creative model to support short-term operations based

on prognostic and health management. This method showed its effectiveness to support operators in the fields of predicting failure and delaying maintenance.

In short, the above design methods showed great potential for solving various barriers to PSS design, and they have already been tested in multiple cases. However, considering that there are three different types of PSS, and this system involves multiple industries, there is still a need for contributions related to practice in more industries. Furthermore, to develop PSS designs to become more suitable and feasible for the current industry, there is also a demand for more studies related to organizational management, knowledge management, and risk management.

### 2.2. Definition of Perturbation and PSS Perturbation

The definition of perturbation was firstly proposed in the field of ecological research, and it is defined as 'An effect; the response of an ecological component or system to disturbance or other ecological processes as indicated by deviations in the values describing the properties of the component or system relative to a specified reference condition; characterized by direction, magnitude, and persistence' (Odum et al. 1979). In the research field, the concept of perturbation usually has the same meaning as the concepts of disturbance and stress, which are usually considered critical sub-concepts of system vulnerability. Thus, in the initial period, tremendous studies used the terms disturbance and stress instead of perturbation, which led to cognitive difficulties arising on the level of terminology (Franz 1981; Edward and Rykiel 1985). In 1996, the concept of disturbance was used in the field of manufacturing, which is defined as 'a disturbance is an unplanned or undesirable state or function of the system'. In general, perturbation refers to any events that change the state of a system into an unwanted state. The initial classification of perturbation was given by Edward and Rykiel (1985), who divided perturbation into two types according to the period of effect. This classification includes two levels, namely, transient and permanent. The explanations of the above two levels are shown below: 1. Transient: a temporary deviation that becomes zero over time with a return to the approximate original steady state. 2. Permanent: deviation that becomes fixed in magnitude over time, leading to a steady state different from the original.

Compared with the general system field, PSS is still lacking enough discussions about perturbation. The only definition was proposed by Estrada and Romero (2016), where a perturbation is any endogenous or exogenous event that modifies the stated PSS operational conditions. Usually, this modification of stated PSS operational conditions refers to an unwanted change. They emphasize that the existence of a perturbation may lead to failure if it is not solved. However, considering that PSS is a system that consists of multiple components, the actual PSS perturbation is even more complex than the explanation of the definition. Wang et al. (2022) propose that there are four vulnerable components of PSS, namely, products, service actors, tasks, and systems. Due to the vulnerability of PSS, the above four components cannot perform a reliable response to a negative event, which leads to the existence of perturbation. In short, a clear expression of PSS perturbation should be a combination of cause and effect, namely, a risky event and its influence on the performance of products, service actors, tasks, and the system. For different components, the effect is also different. For products, the effect refers to weakened availability. For service actors who provide services to customers, the effect refers to the weakened service quality. For tasks, which are the events that are required to be executed to run the PSS, the effect refers to weakened efficiency. For the system, given that PSS is a business system, the effect refers to both weakened efficiency and profitability.

### 2.3. The Chaos of Keywords for Searching Literature Related to PSS Perturbation

According to the provided definition of PSS perturbation in Section 2.2, PSS perturbation is any event that leads to an unwanted change in the performance of PSS. However, more factors need to be considered when researchers are assessing PSS perturbations. PSS is a complex system that involves multiple components. Thus, this description is still

too vague to understand the detail and range of unwanted changes in the field of PSS. To solve this problem, in the study by Wang et al. (2022), a rank of the severity of PSS perturbation was proposed. This classification is based on the theory of Hara et al. (2008), which classified PSSs into products, service actors, tasks, and systems. Service actors are the human wares involved in a service. Products are the hardware involved in the service activity. The task is the service activity, and the system is the combination of all human wares, hard wares, and tasks, which is the product–service system. Based on the above knowledge, Wang et al. (2022) divided PSS perturbation severity into three levels, namely, deviation, disruption, and disaster. Any event that brings about an obvious deterioration in the performance of the vulnerable PSS components could be considered a PSS perturbation. The deviation refers to the situation that the ability of the service actor or the quality of the product is weakened due to internal and external perturbations. At this level, related events do not interrupt the service. Instead, these events lead to a deterioration of perceived quality. For example, due to inadequate training, unqualified service providers provide low-quality services to customers. The disruption refers to the condition that, a specific function of the system cannot stay operational due to one or several unexpected events. For example, due to a traffic jam, the supply of products is delayed. The disaster refers to the condition that the total system cannot stay operational due to a destructive event. Perturbations on the level of disaster usually have a low level of possibility but a high level of loss. This loss is usually unbearable for a PSS; that is, the PSS loses value on the social or economic level for some time. For example, during the pandemic period of COVID-19, shared bike systems lost customers due to concerns about viruses from customers.

Although the above content could support researchers in understanding the features of PSS perturbation, for conducting a literature review, there is a serious lack of effective keywords. To date, limited literature uses 'perturbation', 'stress', or 'disturbance'. This condition does not mean that the current PSS field lacks focus on unwanted change. Indeed, there are other terms that also focus on unwanted changes in PSS performance.

The term operational risk has various explanations, and one of the most popular definitions is 'the direct or indirect loss resulting from inadequate or failed internal processes, people and systems, or external events' (Moosa 2007; Robert Morris Associates et al. 1999). The research by Reim et al. (2016) has paid substantial attention to understanding operational risk, classifying it into behavioral risk, technical risk, and delivery risk. The above risks are considered to pose a threat to the performance of PSS operations. The term barrier is popular in the field of PSS. In this study, the authors argue that both operational barriers and implementation barriers can be equally understood as PSS perturbations. If these operational barriers cannot be overcome, a PSS will not perform as expected or even cannot operate during the operational phase. For example, a sudden technical breakdown would lead to the disruption of the information system of a PSS, which stop the function of communication and monitoring. If the implementation barrier cannot be solved, the unwanted change could appear at the beginning of the PSS operation, which is often caused by the inability to handle the barrier in the implementation phase, which is different from accidental perturbation. For example, customer resistance to the rental culture promoted by PSS has been a major hurdle, leading to a drop in demand, which has prevented business performance from performing at the level expected from the PSS's inception. In short, although both keywords are related to the negative effects of events on PSS performance, there are still some differences. The effect on PSS brought by barriers usually begins from the starting point of the PSS operation due to the inability of the PSS provider. By contrast, operational risk does not have a serious focus on the time issue. It is still unclear whether the above two keywords could effectively contribute to a literature review related to PSS perturbation.

Another related term is 'service paradox'. The concept of service paradox refers to the risk faced by manufacturing firms who want to develop their service businesses (Gebauer et al. 2005). It has been shown that the profitability of a firm is not improved by additional services. The concept of service paradox has raised concern about the success of

the servitization of manufacturing firms. For this concept, there is an interesting feature, in that many events under this concept can also bring unwanted changes to the performance of PSS firms, which shows a relationship with perturbation. For example, for some firms, profitability deviates in a negative direction, which shows the characteristics of a deviation-level perturbation. The transformation is even considered a cause of bankruptcy, which is a disaster-level perturbation for PSS operations (Oliva and Kallenberg 2003; Martinez et al. 2010). Compared with general PSS perturbation, this concept has a strong focus on manufacturing firms.

### 2.4. Gaps to Be Filled

As mentioned in the previous sections, two gaps need to be filled. (1) Previous studies lack direct discussion about PSS perturbations. Various papers tend to use different terms to describe a perturbation or a type of perturbation. There is considerable misunderstanding of the detailed category of PSS perturbation. (2) Although the authors could find useful information related to perturbations from aspects of barriers, failures, and operational risks, the actual effectiveness of this literature review is currently unknown. It is questionable whether various terms could help identify effective perturbations; there is a need to screen out effective perturbations from various similar events.

## 3. Methodology

The result of a literature review is decided by the quality and scope of the involved papers. Thus, a systematic literature review methodology with a well-structured process is required for researchers to grasp the critical knowledge of a large range of reviewed papers (Moher et al. 2009). In this paper, a popular systematic literature review framework called Preferred Reporting Items for Systematic Reviews and Meta-Analysis (PRISMA) was used (Moher et al. 2009). To analyze the data identified through the literature review, this paper adopts a taxonomy methodology proposed by Ackermann et al. (2011) and Ma et al. (2005). To conduct the process of taxonomy building, a group of two Ph.D. students who have experience and knowledge was involved.

### 3.1. The Process of Systematic Literature Review

To effectively identify PSS perturbation, a systematic literature review was carried out. This literature review adapted one of the most famous methodologies of systematic literature review, which is called the PRISMA statement (Moher et al. 2009). In 2019, it was proposed that this methodology could also be applied to the field of engineering design (Lame 2019). In the field of PSS, this methodology was already used by Mahl et al. (2021) and Guzzo et al. (2019), which shows its feasibility. The methodology includes four steps: 1. Record identification, 2. record screening, 3. eligibility assessment, and 4. inclusion. The total process can be seen in the flow chart in Figure 1.

In the phase of record identification, the major issues authors should consider are a database and keywords. In terms of a database, this review adopted an authoritative database, which is called 'Scopus'. Scopus is a famous and reliable academic database that consists of a tremendous amount of literature. To ensure that the result of the literature review is reliable, a replicable review was conducted in the 'Google Scholar' engine.

In terms of keywords, the field term was decided to be 'product service system', which is a relatively developed research field with a tremendous amount of literature. Given that some literature does not use the term 'perturbation' when researching similar events, this study did not only use the keyword 'perturbation'. Instead, this research used 'disturbance' and 'stress' to avoid problems related to similar terms. Furthermore, based on the information in Section 2.2, the concept of perturbation consists of two parts, namely, an event leading to a negative impact and a negative influence on PSS performance during operation. It is highly possible to find knowledge from the papers related to the keywords discussed in Section 2.2. Thus, this paper utilized two sets of combinations of keywords to search for information about perturbation. For the first combination,

this review considered the term 'barrier' as a keyword related to an event leading to a negative influence and considered the term 'failure' as a negative influence. The term 'operational risk', which is used to illustrate the probability of a perturbation to weaken the performance of a PSS operation, was also targeted as a keyword. The terms used in this review are shown below: 'Product-service system' AND 'barrier' OR 'operational risk' OR 'failure' OR 'perturbation' OR 'disturbance' OR 'stress'. For the second combination, this research uses the combination of 'manufacturing' AND 'service paradox'. The utilization of manufacturing aims to prevent reviewing some papers related to the pure service industry. The literature review started on 1 May 2022. In this phase, there were 163 papers screened. Through the authors' identification, 8 works were also selected for this review. The studies of Sakao et al. (2013), Reim et al. (2018), Benedettini et al. (2015), Baines et al. (2009, 2020), Oliva and Kallenberg (2003), Martinez et al. (2017) were recorded as literature from other sources. Thus, there were 171 papers selected as targeted papers.

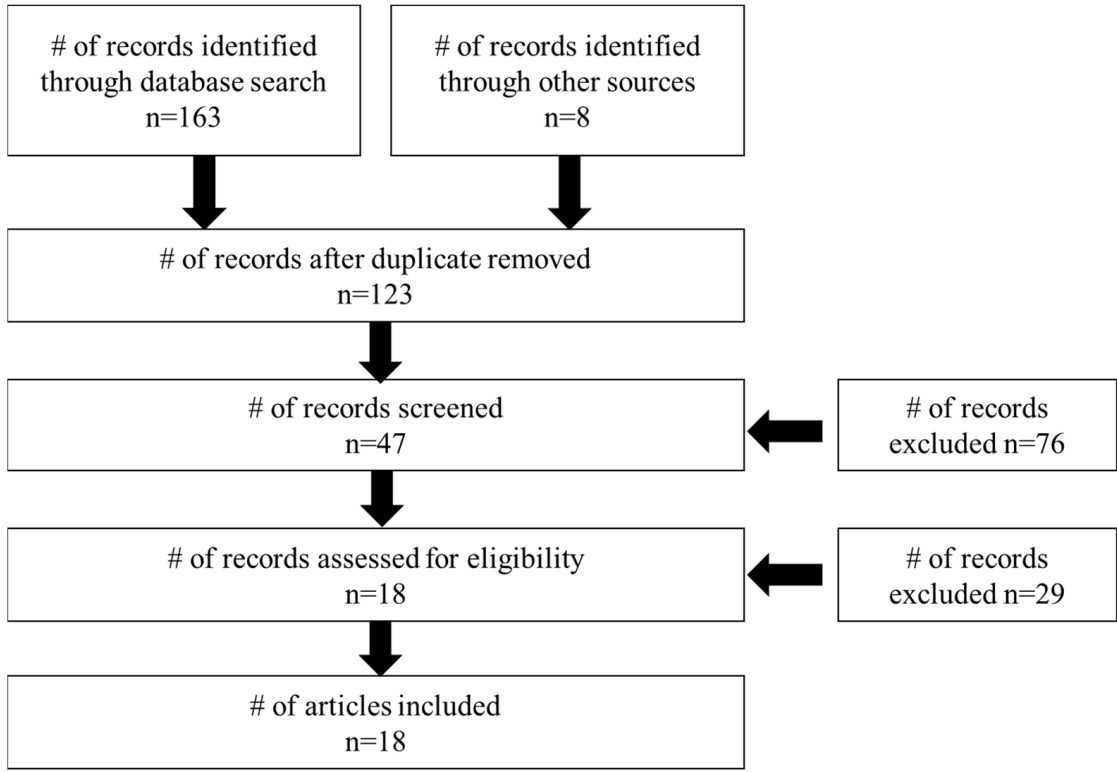

**Figure 1.** The process of the literature review based on Page et al. (2021).

In the phase of record screening, the target of this phase was to exclude all literature that was not related to perturbations of PSS. To achieve this aim, the authors examined the abstracts and conclusions of 171 papers. After this round of examination, the unsuitable topics were removed, which led to 123 excluded papers.

In the phase of eligibility assessment and inclusion, the target was to screen whether all literature was eligible enough to provide effective knowledge. Thus, one principle of assessment was proposed: the keyword of perturbation was not essential to be mentioned in this paper; however, the eligible literature must have focused on risks or events that deteriorated the performance or quality of a product–service system during the operation stage. In addition, an effective item that can describe an event related to PSS perturbation was required. Based on this principle, the authors read 47 left papers, and 29 papers were excluded. As a result, 18 papers were identified as eligible for this systematic literature review and included in the final step. Furthermore, in the process of reviewing, 194 items of events were considered related to PSS perturbations, which could lead to an unwanted change in business performance (see Table 1).

**Table 1.** The list of eligible works.

| Code | Author | Title | The Number of Captured Events Related to Perturbations |
|---|---|---|---|
| 1 | Reim et al. (2016) | Risk management for product-service system operation | 8 |
| 2 | Sakao et al. (2013) | Uncovering benefits and risks of integrated product service offerings—Using a case of technology encapsulation | 4 |
| 3 | Moro et al. (2018) | Barriers to bicycle sharing systems implementation: analysis of two unsuccessful PSS | 25 |
| 4 | Vezzoli et al. (2015) | New design challenges to widely implement 'Sustainable Product–Service Systems' | 11 |
| 5 | Moro et al. (2020) | Product-service systems benefits and barriers: an overview of literature review papers | 26 |
| 6 | Besch (2005) | Product-service systems for office furniture: barriers and opportunities on the European market | 5 |
| 7 | Kuo et al. (2010) | Barrier analysis for product service system using interpretive structural model | 14 |
| 8 | Reim et al. (2018) | Mitigating adverse customer behaviour for product-service system provision: An agency theory perspective | 3 |
| 9 | Inagaki et al. (2022) | Extracting the relationship between product-service system features and their implementation barriers based on a literature review | 21 |
| 10 | Benedettini et al. (2015) | Why do servitized firms fail? A risk-based explanation | 3 |
| 11 | Baines et al. (2020) | Framing the servitization transformation process: A model to understand and facilitate the servitization journey | 3 |
| 12 | Coreynen et al. (2017) | Boosting servitization through digitization: Pathways and dynamic resource configurations for manufacturers. | 4 |
| 13 | Martinez et al. (2017) | Exploring the journey to services | 5 |
| 14 | Oliva and Kallenberg (2003) | Managing the transition from products to services | 6 |
| 15 | Baines et al. (2009) | The servitization of manufacturing: A review of literature and reflection on future challenges | 6 |
| 16 | de Jesus Pacheco et al. (2019a) | Overcoming barriers towards Sustainable Product-Service Systems in Small and Medium-sized enterprises | 12 |
| 17 | Kamal et al. (2020) | Servitization implementation in the manufacturing organisations: Classification of strategies, definitions, benefits and challenges | 34 |
| 18 | Dmitrijeva et al. (2022) | Paradoxes in servitization: A processual perspective | 4 |

### 3.2. Successive Refinement

In this section, a procedure of successive refinement to categorize the PSS perturbations' taxonomies is provided. To ensure that the choice of terms was clear and precise, the method of Ma et al. (2005) and Ackermann et al. (2011) was adopted. This method is an effective taxonomy-building method and has already been utilized in the field of the taxonomy of risk. The three steps of this method are going to be introduced in the following sections. To fulfill the following steps, a group of two Ph.D. students who have considerable knowledge about PSS was required to conduct the procedures.

### 3.2.1. Item Reduction

The initial identification of items related to PSS perturbation showed that there were 119 items. Given that there was a certain level of possibility that some items describe the same events related to perturbations, it was essential to reduce the number of similar items and include them in one item. For example, 'rejection of change by internal personnel' and 'resistance towards PSS inside firm' represent the same meaning. Thus, the authors needed to remove the duplicates.

Given that this literature review was also based on the keywords 'service paradox', 'barrier', and 'operational risk', there were some items that could not be regarded as a perturbation. For example, the term 'rebound effect' refers to a side effect of some solutions on PSS. In this case, the rebound effect is more like a type of influence rather than an event. In addition, some events were not directly linked to the deterioration in a PSS's business performance. For example, the item 'complex supply chain' does not directly worsen the business performance of a PSS. What really exacerbates it is the lack of capacity of PSS providers and suppliers and the lack of links between providers and suppliers. Therefore, in this step, reducing an item followed the following principles:

1. The item should show a clear relationship between the cause and its unwanted influence on PSS.
2. Terms with the same meaning should be integrated into one representative term.

Based on the above principles of item reduction, the group identified 43 effective items, and 151 items were reduced.

### 3.2.2. Initial Grouping

Every individual in the group was required to group items of PSS perturbation into different classifications. As for category names, team members needed to summarize similar names into an entry recognized by all. For redundant or useless category names, team members needed to delete them.

Moreover, to shorten the number of items, the team was also asked to group some items with similarities into a single item. For example, 'Resistance to established local habits', 'Rejection of change by internal personnel', and 'Resistance to consumption without possession' all demonstrate stakeholders' resistance to PSS features, but in different ways. Thus, the term 'stakeholders' resistance to change' could take place of the above three items. Based on the above rules, 24 items were included as effective ones.

For the selection of categories, the group made a group discussion about the features of the involved items. First, the members identified items that mentioned their belonging categories. For example, for behavioral perturbation, all items contained the word 'behavior', which could be identified directly. Second, for the remaining items, the members checked their meanings and explanations in related literature and selected a category for them. Based on the result of the above steps, six categories were set. For items related to behavior, the category was behavioral perturbation. For items related to a resource that the firm needs to obtain from the market, the category was competence perturbation. For items related to the lack of competence to overtake the function of PSS, the category was competence perturbation. For items related to problems related to the structure of the organization, the category was organizational perturbation. For items related to the external environment, including legal, market, and macro-economy, the category was environmental perturbation. For items related to social culture and attitude, the category was social perturbation.

### 3.2.3. Regrouping

In this step, the members of the group were asked to regroup the items. The authors put all items into every category and individuals were required to check their similarities and differences. Furthermore, it was critical to check whether an item was suitable for multiple groups. For example, if the categories of 'attitude perturbation' and 'social perturbation' were set, then it would be extremely difficult to decide the position of the term 'resistance towards PSS novelty'.

Furthermore, this step was also required to examine the overlap between various items. Individuals in the group needed to make sure that every item only belonged to a single category. The outcome of the grouping had to be agreed upon by all members such as that there was no other way to classify these items. As a result, 25 items were categorized into 6 groups, namely, behavioral, resource, competence, organizational, environmental, and social perturbation. Regarding this outcome, the members of the group had no divergence about it. The details of the categories and their related items are shown in Table 2.

**Table 2.** The details of the categories and their related items of PSS perturbations.

| Category | Involved Items | Involved Papers |
| --- | --- | --- |
| Behavioral perturbation | Careless behavioral | Reim et al. (2016, 2018) |
| | Opportunistic behavioral | Reim et al. (2016, 2018); Moro et al. (2020) |
| | Reverse selection | Reim et al. (2016, 2018) |
| Resource perturbation | Need for high initial investment | Besch (2005); Coreynen et al. (2017); Moro et al. (2020) |
| | Lack of personnel | Baines et al. (2009); Kamal et al. (2020); Kuo et al. (2010); Moro et al. (2018); Oliva and Kallenberg (2003); Vezzoli et al. (2015) |
| | Lack of material and resource | Baines et al. (2020); Benedettini et al. (2015); Coreynen et al. (2017); de Jesus Pacheco et al. (2019a); Reim et al. (2016) |
| | Lack of infrastructure | Inagaki et al. (2022); Oliva and Kallenberg (2003); Vezzoli et al. (2015) |
| Competence perturbation | Lack of experience in service design and offering | Baines et al. (2009); Benedettini et al. (2015); Moro et al. (2018, 2020); Vezzoli et al. (2015) |
| | The difficulty of controlling and managing materials | Kuo et al. (2010) |
| | The difficulty of controlling costs/accounting | Coreynen et al. (2017); Moro et al. (2018, 2020) |
| | Difficulty in performing reverse logistics | Besch (2005); Kuo et al. (2010); Moro et al. (2018, 2020) |
| | Lack of technical competence | Kamal et al. (2020); Kuo et al. (2010); Martinez et al. (2017); Reim et al. (2016); Sakao et al. (2013) |
| | Lack of training competence | Kamal et al. (2020); Kuo et al. (2010); Martinez et al. (2017); Vezzoli et al. (2015) |
| Environmental perturbation | Prerequisites change in the case of a result-oriented contract | Sakao et al. (2013) |
| | Lack of legal support | Moro et al. (2020); Vezzoli et al. (2015) |
| | Competitive market | Baines et al. (2009); Coreynen et al. (2017); Moro et al. (2020) |
| | The economic downturn and industry recession | Moro et al. (2020); Benedettini et al. (2015) |
| Organizational perturbation | Lack of stakeholders' engagement | Besch (2005); Kamal et al. (2020); Kuo et al. (2010); Martinez et al. (2017); Moro et al. (2018, 2020) |
| | Low transparency and exchange of information between partners | Moro et al. (2018) |
| | Inappropriate organizational structure | Baines et al. (2020); Besch (2005); de Jesus Pacheco et al. (2019a); Reim et al. (2016) |
| | Wrong strategy | Baines et al. (2009); Oliva and Kallenberg (2003) |
| | Internal conflicts between sales and service areas | Coreynen et al. (2017); Dmitrijeva et al. (2022) |

**Table 2.** *Cont.*

| Category | Involved Items | Involved Papers |
|---|---|---|
| Social perturbation | Resistance to change by stakeholders | Baines et al. (2009); Coreynen et al. (2017); de Jesus Pacheco et al. (2019a); Kamal et al. (2020); Kuo et al. (2010); Moro et al. (2018, 2020); Oliva and Kallenberg (2003); Vezzoli et al. (2015); Dmitrijeva et al. (2022) |
| | Lack of acceptance toward the design of products and services | Baines et al. (2009); Besch (2005); Coreynen et al. (2017); Moro et al. (2018) |
| | Lack of awareness related to PSS | Moro et al. (2020) |

## 4. Taxonomy of PSS Perturbation

*4.1. Introduction*

In Section 3.2.3, the concept of PSS perturbations was classified into six major categories, namely, behavioral, resource, competence, organizational, environmental, and social perturbation. Based on the above categories, the authors built a taxonomy for PSS perturbation. To enable people to understand the meaning and knowledge of these categories, a detailed explanation is provided as follows.

*4.2. The Explanation of the Categories*

4.2.1. Behavioral Perturbation

The item of behavioral perturbation refers to events related to a customer's adverse behavior, which could destroy products or reduce a product's life. During the influence period of this type of perturbation, the maintenance efficiency and product availability seriously deteriorate.

The problem of customer adverse behavior is a typical PSS perturbation, which has been focused on in the research of Reim et al. (2018) and mentioned by Reim et al. (2016), Sakao et al. (2013), and Moro et al. (2020). Compared with the traditional manufacturing industry, PSS companies often need to provide customers with a longer period of business, such as maintenance and leasing rather than simply selling products (Reim et al. 2018). Therefore, customers have more possibilities and time to damage products, and companies must consume a lot of resources for maintenance and remanufacturing. According to Reim et al. (2016, 2018), adverse customer behaviors can be observed from three perspectives, namely, careless behavior, opportunistic behavior, and reverse selection. Careless behavior refers to damage to products caused by customers when they do not pay enough attention during operation. Opportunistic behavior and reverse selection refer to intentional destruction or behavior. For PSS, especially user-oriented and result-oriented PSSs, there is no available regulation that could punish customers if they lead to a breakdown of products. It seems that PSS providers can only expect that a high-level morality in customers could prevent such types of events. This general condition promotes the motivation of immoral behavior, including intentional destruction and earning money through a renting contract. It has been reported that some customers even tend to use vulnerable machines to obtain compensation (Reim et al. 2016).

4.2.2. Resource Perturbation

The item of resource perturbation refers to any events related to a lack of financial resources, human resources, or material and natural resources, which weaken the performance of a PSS. A lack of resources could lead to difficulty for PSS firms to manufacture products and operate the system, which causes a further negative influence on the efficiency of the system.

The lack of financial resources is usually given high-level importance for the stable operation of PSS. For user-oriented PSS, these PSSs often require a large initial investment (Moro et al. 2018; de Jesus Pacheco et al. 2019a) and might have a high cost (Moro et al. 2020). This makes PSS firms have a strong requirement for stable cash flow to stay operational.

A long-term lack of financial resources would lead to the bankruptcy of PSS firms. A lack of personnel has also been cited as a risk factor by multiple kinds of research (Moro et al. 2018; Vezzoli et al. 2015; Kuo et al. 2010). There is also a lack of infrastructure to support the operation of PSS (Vezzoli et al. 2015; Inagaki et al. 2022). Compared with manufacturing firms, the daily operation of PSS requires more staff to support the provided service and products (Kamal et al. 2020; Baines et al. 2020). A lack of experienced service actors would lead to poor service quality and low efficiency. The number of available resources is considered a critical factor for PSS offerings and operations, which can lead to the condition that no material is available for machine production (Reim et al. 2016; Benedettini et al. 2015; Baines et al. 2020). Furthermore, the lack of infrastructure has also been mentioned (Oliva and Kallenberg 2003; Kamal et al. 2020). For some manufacturing firms, a major barrier to servitization is the lack of infrastructure related to IT and services (Oliva and Kallenberg 2003; Inagaki et al. 2022).

### 4.2.3. Competence Perturbation

The item of competence perturbation refers to perturbations caused by a lack of a specific competence in the PSS to conduct its duty in any aspect, including design, accounting, finance, monitoring, and management. This type of perturbation involves two kinds of events:

1. Accidental events: When an accident happens, the competence of the service provider plays an important role in mitigating the loss and restoring the operation.
2. Known barriers: This type of event is known as a barrier since the design and implementation stage by PSS providers. However, due to a lack of specific competence, the loss is still caused.

Based on the result of the literature review, technical competence is considered an important issue, especially when related to monitoring and IT (Kamal et al. 2020; Martinez et al. 2017). A lack of monitoring techniques is believed to result in product malfunction, which reduces the availability of products and the efficiency of the maintenance system (Reim et al. 2016; Sakao et al. 2013). There is also some focus on other types of competence, including a lack of experience in service design and offerings (Baines et al. 2009; Benedettini et al. 2015; Moro et al. 2020), lack of control and management material (Kuo et al. 2010), difficulty in accounting (Coreynen et al. 2017; Moro et al. 2018; Inagaki et al. 2022), and difficulty in performing logistics and reverse logistics (Besch 2005; Kuo et al. 2010). It has been shown that competence in assessing risk and cost deserves more attention, especially for result-oriented PSS. To ensure that sharing the machine could provide more profits for the firm, the choice of material and energy for saving waste plays a crucial role in the long-term benefit of the machine (Coreynen et al. 2017). There is also a lack of training related to specific skills including communication and IT skills (Kuo et al. 2010; Martinez et al. 2017; Kamal et al. 2020).

### 4.2.4. Organizational Perturbation

Organizational perturbation mainly refers to any events that originate from the organizational structure that hinder the operation and development of the system and collaboration among different stakeholders. The low efficiency and profitability of firms is a usual unwanted change caused by this perturbation.

The prominent sub-perturbation under this group is the low engagement of stakeholders. Multiple works propose that PSS providers might face difficulty in operation due to a lack of support from senior management (Kuo et al. 2010; Martinez et al. 2017; Baines et al. 2020; Kamal et al. 2020). The sudden loss of senior support due to personal reasons has also been recorded as an accidental disaster for manufacturing firms to develop themselves into PSS providers (Baines et al. 2020). Moro et al. (2020) and Besch (2005) illustrated that a PSS's cost and efficiency are influenced by the engagement of supply chain and logistics operators. Moro et al. (2018) proposed that a low-level engagement from

the implementation team is a major barrier to a shared bike system being implemented and operated.

Furthermore, the low transparency and exchange of information between partners have also been proposed as organizational problems that could deteriorate the efficiency of a system (Moro et al. 2018). Some firms tend to separate the departments of services and sales; this is regarded as a cause of internal tension, which further reduces efficiency (Dmitrijeva et al. 2022). The inappropriate structure is illustrated as a major source of operational risk, which could weaken the performance of the system (Reim et al. 2016). In the case of shared furniture in North Europe, Besch (2005) found that a decentralized structure is more suitable for PSS when the shipment fee is high. Conversely, a centralized structure can lead to a dramatic increase in the cost of logistics, leading to poorer economic performance. Furthermore, internal organizational conflicts between sales and service areas are believed to be risky in the operation phase. Furthermore, a lack of appropriate organizational strategy has also been given importance. PSS firms have been shown to require organizational readiness, namely, robust processes and products (Baines et al. 2020). For the different stages of servitization, a sustainable and reliable strategy is important for winning the market (Oliva and Kallenberg 2003; Baines et al. 2020).

### 4.2.5. Environmental Perturbation

Environmental perturbation is used to refer to any event that leads to a change in the legal and economic environment of a PSS that then changes the prerequisites of the contract. This change is usually unwanted, which could make the firm fail to fulfill the commitment and even disrupt the operation of the system.

In discussions about environmental perturbation, the topic mentioned most of the time is legal environmental problems. Legal support, especially related to laws that support the dissemination of shared use, is considered a key policy for operating a PSS (Hannon et al. 2015; Vezzoli et al. 2015; Moro et al. 2020). For many PSS firms, there is a high expectation for government actions to propose an educational strategy that can guide customers to become familiar with this novel renting business mode (Kuo et al. 2010). Furthermore, a policy that encourages customers to buy products is also regarded as a major barrier that hinders the promotion of user-oriented PSS, which reduces the demand for PSS.

Furthermore, despite the limited discussion, the unwanted change in the economic environment, namely, the competitive market and fluctuation of prices of materials, is also worth attention. A lack of profitability and a restricted market is proposed as a major barrier that hinders the economic performance of PSS. The competitors' imitative actors are considered major threats for PSS firms to survive (Baines et al. 2009; Coreynen et al. 2017). In terms of external economic environments, economic downturn and industry recession, which lead to a disrupted market, are considered major causes of the breakdown of PSS firms (Benedettini et al. 2015; Moro et al. 2020).

### 4.2.6. Social Perturbation

The term social perturbation describes any adverse attitude toward PSS on a social level that can weaken the acceptance, trust, and satisfaction toward this novel system. These kinds of events do not have a direct and physical influence on the products. Instead, they show their threat on a social and psychological level.

Generally, adverse social attitudes toward PSS can be classified into three major types, namely, resistance to change, a lack of acceptance toward the design of products and services, and a lack of awareness related to PSS.

Compared with the other two types, resistance to change is given higher importance according to discussions in the majority of the reviewed papers. This resistance is believed to be related to the operational challenge inside PSSs, which can disrupt the operation and weaken the service performance. Internal rejection of PSS novelty is a critical problem, which requires some staff to learn how to provide services and understand the value proposition of PSS (Oliva and Kallenberg 2003; Baines et al. 2009; Kuo et al. 2010; Moro

et al. 2018; de Jesus Pacheco et al. 2019a; Kamal et al. 2020; Dmitrijeva et al. 2022). There is strong resistance from manufacturing staff to learning knowledge about services. This perturbation happens in the initial period of operation and implementation, as internal personnel do not have enough knowledge and understanding in this stage. For example, for PSS firms to implement this system in a new context, resistance to established local habits has been proposed as a prominent challenge (Moro et al. 2020). For customers, there is also a problem with resistance toward consumption without possession (Moro et al. 2018), and they have also been found to be sensitive to being monitored by PSS providers when they are renting or using a product (Vezzoli et al. 2015). The above resistance issues could lead to lower efficiency and low interest in PSS, which further reduces the demand of customers.

The lack of acceptance for the design of products and services is considered a prominent type of perturbation on the performance of demand and satisfaction. Some are reluctant to change their habits about purchasing products (Coreynen et al. 2017). According to the finding of the case study of a shared furniture business in North Europe, customers were found to be sensitive to the price of products and services. Furthermore, for the design of products that are used for sharing, the element of fashion is given a high-level focus. Customers present a preference for fashionable and new products, which means that there is a high probability for them to reject using shared furniture or reject continuing a contract after finishing one (Besch 2005). Consumers also show a strong desire to have technologically up-to-date products (Moro et al. 2018).

Furthermore, for the issue of awareness related to PSS, customers are considered a major stakeholder that holds this attitude. It has been shown that users have a hard time perceiving the economic advantage of PSS. In addition, low awareness of environmental impacts reduces the value of PSS for potential customers. Furthermore, since service is intangible, it is difficult for customers to grasp the intangible value of the additional services of PSS (Moro et al. 2020).

## 5. Discussion

### 5.1. The Categories of PSS Perturbation

In this research, six categories of PSS perturbations are included in a taxonomy, namely, behavioral, resource, organizational, social, competence, and environmental perturbation. The items used for naming the above six categories are considered the major source of risky events, which could result in an unwanted change in the performance of the components of PSS. A brief explanation is provided as follows.

Behavioral perturbation refers to any customer's adverse behavior that leads to product breakdown. Since customers are usually also the users of machines and products, they could have a direct and negative influence on the products. This would also deteriorate the efficiency of maintenance. In this type of perturbation, user-oriented PSS has received tremendous attention. Much evidence shows that user-oriented PSS cannot have good control over the random behavior of customers (Reim et al. 2016, 2018). By contrast, there are limited specific discussions about product-oriented and result-oriented PSSs in terms of this category.

Resource perturbation refers to any shortage of resources, including investment, labor resources, materials, and infrastructure, that leads to the disruption of the task operation (Mont 2004; Reim et al. 2016) and even the bankruptcy of PSS firms (Vezzoli et al. 2015; Moro et al. 2018; Benedettini et al. 2015).

Organizational perturbation refers to any structural problem that originates from the organization that reduces efficiency. This category includes the low transparency and exchange of information between partners (Mont 2002; Moro et al. 2018) and low-level involvement from stakeholders, including senior managers (Kuo et al. 2010), logistics providers, suppliers (Besch 2005; Moro et al. 2020), and implementation teams (Moro et al. 2018).

Social perturbation refers to any adverse social attitude from various stakeholders, including customers, managers, staff, government, and the local community, that reduces

the acceptance and satisfaction of the above stakeholders. The adverse social attitude can be mainly divided into two types, namely, insensitivity toward sharing or renting (Kuo et al. 2010; Vezzoli et al. 2015; Moro et al. 2018), resistance toward PSS novelty (Besch 2005), and lack of awareness related to PSS (Moro et al. 2020).

Competence perturbation refers to any lack of specific competence to overtake the basic function of PSS and the ability of service actors to maintain the system against accidental events, which weakens the performance of tasks and the availability of products. For the basic function of PSS, service offering and design, accounting, technology, and performing reverse logistics were mentioned by the reviewed literature (Besch 2005; Kuo et al. 2010; Benedettini et al. 2015; Moro et al. 2018; Inagaki et al. 2022). For maintaining the system, the discussion remains limited, which was only discussed by Moro et al. (2020).

Environmental perturbation refers to any lack of environmental prerequisites and support from the legal environment, market, and micro-economy that reduce acceptance and profitability. Among various types of environments, it is notable that the legal environment is given a high-level focus. Previous researchers have proposed that a lack of legal support, including laws and regulations about sharing dissemination, is the major cause of the unacceptance of user-oriented PSS (Kuo et al. 2010; Hannon et al. 2015; Vezzoli et al. 2015; Moro et al. 2020).

### 5.2. The Features of PSS Perturbation

According to the taxonomy, there are several features of PSS perturbations that are different from operational risk and barriers. An introduction to these concepts is shown in the following. First, PSS perturbation can bring an unwanted change to the initial performance of PSS. Indeed, the vulnerable condition could start from the first moment of operation. It is not wise to consider a weakened performance a normal condition only because it is born to be like that. This feature is usually not mentioned in the study of operational risk, which has a strong focus on accidental events. This feature is well explained by parts of items of social, resource, organizational, environmental, and competence perturbation. For social and environmental perturbations, the acceptance of PSS is weakened due to a lack of support from the government and culture. For example, the terms 'resistance to consumption without possession' and 'lack of appropriate policies for dissemination of shared use' have a direct relationship with a reduction in customers' acceptance of a new PSS (Moro et al. 2018, 2020; Vezzoli et al. 2015). For competence, resource, and organizational perturbations, the terms 'lack of experience in service design and offering', 'low transparency and exchange of information between partners', and 'lack of infrastructure show that efficiency could be threatened from the start of a PSS operation. The term 'need for high initial investment' shows that there is a disaster-level perturbation for PSS to survive regarding the risk of bankruptcy in the initial period.

Second, PSS perturbation is not only an anticipated event but also an accidental event with high-level uncertainty. PSS researchers should not be bound by the logic that perturbation can only appear as a form of an anticipated event. It has been found that terms related to barriers are usually known events, which would definitely hinder the operation of a PSS if there is no solution. For example, the term 'need for initial investment' is an essential requirement for a PSS to survive or perform a normal performance. In other words, events related to barriers are usually established events. They will appear in a predictable situation, and they will have an adverse impact on PSS without effective measures. By contrast, terms related to operational risk show a strong focus on accidental events, of which it is uncertain whether they will happen or not. Behavioral perturbation is a typical accidental perturbation. When customers damage products, the occurrence time and severity of the damage are difficult to find and be assessed in advance. The occurrence of events contained in this kind of perturbation is random, and a solution has not been established.

In short, PSS perturbation is an integration of any features of barriers and operational risk. It is an accidental or anticipated event that can bring an unwanted change to the performance of a PSS during the whole operation stage.

### 5.3. A Guideline for Selecting Suitable Terms for Researching PSS Vulnerability and Perturbation

When reviewing literature related to PSS perturbation, there are the following barriers: (1) To date, there is a limited number of papers with the keywords 'perturbation', 'disturbance', and 'stress' in the field of PSS. The existing papers are still at a conceptual level. (2) Although papers with the keywords 'operational risk', 'service paradox', and 'barrier' provide considerable useful information, for some categories of perturbation, there is a low-level agreement between the two types of papers. Some categories of perturbation are overlooked by a single type of paper. To solve the above chaos related to multiple terms, there is a need to provide a guideline for future researchers who are interested in PSS perturbation and vulnerability. The details of the guideline are as follows:

Researchers should not stick to terms such as disturbance, perturbation, and stress. In fact, perturbation is a subsidiary concept of vulnerability, and the core purpose of studying perturbation is to make PSSs insensitive to unwanted changes, which aims to achieve a robust PSS (Wang et al. 2022). For perturbation cognition, the most important criterion is whether any event can cause unwanted changes in PSS performance, not whether the article uses the most relevant keywords. Given that there is a lack of focus on PSS vulnerability and perturbation, identifying potential events that could bring unwanted change from the fields of operational risk, service paradox, and barriers is recommended.

For perturbation identification, the relationship between events and their impacts on PSS performance must be checked. According to the literature review and the finding of Section 5.2, studies related to the terms 'operational risk', 'service paradox', and 'barrier' have limitations due to a focus on different life cycle stages. Papers related to operational risk have a strong focus on the operation stage, which overlooks the stage of implementation and recycling. By contrast, papers related to barriers and the service paradox have more focus on the other stages. However, in papers related to barriers and the service paradox, the authors found that many terms were too broad or lacked a link to unwanted change. For example, in the study of Moro et al. (2020), the rebound effect was considered a hindrance. When this term was evaluated, the group members considered it only an effect and the presence of the event was lacking. Thus, there is a strong need for researchers to make a careful assessment of the relationship inside every term during identification.

### 5.4. Practical Usefulness for Future Research

For future research, this taxonomy is expected to have two practical uses. First, this research is an effective reference for researchers who have an interest in PSS vulnerability and robustness. For researchers who want to mitigate vulnerability and achieve robustness, the most important matter is to understand the scope of events that lead to unwanted changes in performance. Furthermore, this taxonomy has provided a detailed introduction to how these perturbations form and how they lead to a negative impact on PSSs. Thus, a taxonomy of perturbation could provide considerably useful knowledge and information for researchers to study and exchange information.

Second, this research is also expected to support some robust design methods and vulnerability analysis methods for database building. The Taguchi method, the most famous, robust design method, is a good example of illustrating how this taxonomy has practical uses. For the Taguchi method, there are four key parameters, namely, signal factor, control factor, noise factor, and response factors. According to Taguchi (1987), the signal and response factors refer to the expected performance and actual performance of products. The control factor refers to any elements that could improve the performance of the signal factor. The noise factor refers to any uncontrollable factor that leads to unwanted change. In the Taguchi method, researchers need to ensure that the performance could be improved by control factors and be insensitive to noise factors. Thus, it is extremely

important for designers to identify critical noise factors. For previous researchers, the most difficult issue was that designers did not have a clear understanding of the scope of noise factors. Indeed, companies usually do not prepare a database for the cause of the failure. Instead, they just collect the loss of events (Creveling et al. 2002). To date, there is no guidance or support for designers to collect data productively. Given this difficulty, there is a need for PSS designers to find an effective way to identify potential noise factors and build a database. In fact, the knowledge of PSS perturbation taxonomy could provide support. It is worth noting that there is a large extent of similarities between the definition of noise factor and perturbation. The only difference is that noise factor identification requires the consideration of uncontrollability, which is lacking for perturbation. For designers who want to use the Taguchi method, they need to consider whether current resources and competence can solve this perturbation. If the answer is no, then this perturbation is noise. Thus, the noise factor can be regarded as a special type of perturbation, which is costly to control or not controllable for PSS firms. Therefore, a taxonomy of perturbation could narrow the scope of potential perturbations by guiding designers to identify potential perturbations in the range of the six categories. After that, they need to assess the uncontrollability of this event and decide whether it is a noise factor.

### 5.5. Theoretical Significance

In this paper, the authors were devoted to identifying existing and known events that could lead to unwanted changes in PSS performance, namely, PSS perturbations. This is a meaningful theoretical supplement for the knowledge gap in the field of robust PSS design, which is a promising direction to design a sustainable and reliable PSS. According to Taguchi (1986) and Hasenkamp et al. (2009), a robust design aims to enable the system to become insensitive to internal and external noise. The key aspect of realizing robust PSS is that researchers and designers must have a high-level understanding to design components of the system that will become insensitive to various types of noise. However, before this research, the field of PSS design and development did not provide a holistic introduction to events that could be considered perturbations. Indeed, based on the outcome of the systematic literature review, we found that papers with the keywords 'failure', 'operational risk', 'barrier', and 'service paradox' do not have the same focus on multiple terms. This phenomenon is especially obvious in papers related to operational risk and the risk of service offering. The papers by Reim et al. (2016, 2018) and Sakao et al. (2013) show a strong preference for accidental events, including customer behavior and prerequisite changes. In contrast, papers related to the service paradox and barriers are more focused on the implementation stage. The perturbations related to the above two types of paper show a logic where 'if you don't find a solution, these events will show their destruction quickly in the implementation stage'. Thus, this research provides theoretical knowledge and integrates events that bring unwanted changes in various aspects. Furthermore, this paper also made a response to the request of Wang et al. (2022), who noted that there is a lack of further understanding of vulnerability. As a sub-concept of vulnerability, a well-structured taxonomy of perturbation could provide tremendous knowledge about vulnerability.

### 5.6. Future Research Direction

The findings of this study show that currently, PSS is facing multiple categories of perturbation, which means that mitigating vulnerability is a complex and difficult task. According to Wang et al. (2022), the ultimate goal of eliminating vulnerability in PSS is to achieve a robust PSS, which requires PSS designers to design PSSs that are insensitive to perturbations of various categories. According to Arvidsson and Gremyr (2008), robust design means systematic efforts to achieve insensitivity to noise factors, which is a type of perturbation with high-level sensitivity (see Section 5.2). These efforts are based on an awareness of variation. However, the existing research has not made an in-depth analysis of the sensitivity of different types of PSSs toward various perturbations. There is a gap in the understanding of the severity of perturbations. Furthermore, to date, although some

studies have contributed to eliminating the severity of failure in PSS redesigns (Kimita et al. 2018; Mahl et al. 2021), current researchers have not proposed an effective method to reduce the sensitivity of systems toward various noise factors in the stage of design. There are still no existing methods that have been proven to be suitable for improving the insensitivity of PSS. Considering that PSS is a complex system that consists of multiple products and services that fulfill the requirements of customers (Goedkoop et al. 1999), designers need to consider further variables caused by additional services. For example, the Variation Mode and Effect Analysis (VMEA) method is a classic method for reducing sensitivity in the field of robust product design (Johansson et al. 2006; Hasenkamp et al. 2009). This method provides a framework to assess and mitigate potential noise factors based on the key product characteristics (KPCs) of products in the design stage. When conducting this method in the field of PSS, analysts are required to overtake further complex works related to identifying service characteristics and PSS characteristics. For the development of a robust PSS, it is essential to propose a new method or modified method to assess and reduce the sensitivity of the PSS toward noise factors based on the features of the PSS.

Furthermore, to mitigate perturbation and noise factors, managers and designers require a reliable management guideline, which is currently not well constructed. According to Reim et al. (2016), for PSS, especially user-oriented PSS, traditional risk management solutions, namely, risk avoidance, risk share, and risk delivery, do not work properly. It has been proposed that the special orientation of the sharing mode requires the existence of risk to share the ownership of products instead of selling them. Accordingly, for future research, there is also a need to propose a set of mature risk management guidelines to mitigate PSS perturbation. Furthermore, it is worth noting that the existing literature might not provide comprehensive information about PSS perturbation. A limited understanding of PSS perturbation could lead to unwanted loss when an unfamiliar perturbation occurs. Thus, more case studies of real PSSs and interviews with PSS professionals are required. In developing organizational studies about vulnerability and perturbation, there are also some existing problems. Martinez et al. (2017) pointed out that in servitization firms, there is a general tendency not to document lessons learned from successes and failures. Thus, it is also essential to find a method to enable PSS managers and staff to study the experience of previous cases. A systematic guideline about knowledge of vulnerability mitigation is required.

## 6. Conclusions

The problem of vulnerability has become a critical issue for businesses to survive in a competitive market. Given that the possibility and severity of perturbations are usually key indicators to assess whether a system is robust or not, how to reduce the possibility or severity of a perturbation is extremely meaningful in realizing a robust PSS. However, to date, there is still a lack of theoretical research that categorizes and defines various perturbations in the context of PSS as support for vulnerability mitigation. Thus, there is a need to grasp detailed knowledge about PSS perturbation by building a taxonomy of PSS perturbations. To achieve this aim, this paper adopted a systematic literature review as a method to find knowledge about PSS perturbation. Considering there are limited findings about the keyword 'perturbation', this research chose 'operational risk', 'service paradox', and 'barrier', which have the potential to contribute knowledge related to perturbation, as options for keywords. The literature review targeted 171 papers published between 2000 and 2022 as potentially useful papers in the first round of identification. After the identification, 18 papers were identified as effective ones that could provide support for taxonomy building. As a consequence, this research utilized information exacted from the literature review to build a taxonomy of PSS perturbation based on the opinions of two Ph.D. students who are experienced and knowledgeable in the field of PSS. The findings are as follows.

Through the literature review, we found that papers related to operational risk and barriers can also provide a tremendous amount of knowledge in understanding perturba-

tion. Furthermore, we found that perturbation has an obvious and negative influence on the performance of PSSs in the initial period of operation. This could effectively fill in the research gap where vulnerability and perturbation in the field of PSS have been overlooked long-term. In this paper, by building a taxonomy of PSS perturbation, six major categories were defined, namely, behavioral, competence, environmental, social, resource, and organizational perturbations. They are believed to be the major sources of unwanted changes in the PSS context. It is shown that the efficiency, satisfaction of customers, and acceptance of PSS novelty could be deteriorated by the above six types of perturbations. Furthermore, this taxonomy has shown two features of PSS perturbation. First, PSS perturbation can bring unwanted change to the initial performance of a PSS. Second, PSS perturbation is not only an anticipated event but also an accidental event.

Furthermore, this research also discussed its theoretical meaning and practical usefulness. To solve the problem of limited findings related to the keyword 'perturbation' and mitigate the chaos of multiple keywords, the authors proposed a guideline for researchers who are interested in the vulnerability of PSSs. In terms of practical usefulness, this paper should be regarded as an effective reference for research related to the robustness and vulnerability of PSS, given that the general field of robust design requires designers to identify effective noise factors, which is usually a difficult task. The authors also proposed that knowledge of this taxonomy could support researchers and designers of PSSs to build a database for noise factors.

Although the findings of this paper could help researchers further understand perturbation in PSSs, there are undeniable limitations. Firstly, the current lack of attention to vulnerability and perturbation in the PSS field makes the findings of the literature review likely to be one-sided. In fact, PSS only has a development history of more than 20 years, and researchers have performed a lot of research on novelty on the conceptual level, but the existing research usually lacks considerations of robustness and reliability, which are attributes related to the stable operation of a PSS. In the current PSS field, only Reim et al. (2016, 2018) and Sakao et al. (2013) have analyzed risks in terms of PSS operation and offerings. Although these studies have greatly contributed to the risk management and robust design of PSSs, these findings are often not comprehensive due to the limited number of cases. Secondly, although this study supplements perturbation cognition via the perspective of barriers, some barriers may not be the root cause of the perturbation. Human error is possible due to the limitation of the methodology and lack of experience. More experienced researchers and managers are required to join in the process of successive refinement. Thirdly, this paper only selected Google Scholar and Scopus as databases; there are also other available databases for literature reviews.

**Author Contributions:** Conceptualization, H.W.; methodology, Y.M. and H.W.; software, H.W.; validation, H.W., Y.M., Y.T., S.A. and Y.S.; formal analysis, H.W. and S.A.; data curation, H.W.; writing-original draft preparation, H.W.; writing-review and editing, H.W., Y.M., Y.T., S.A. and Y.S.; supervision, Y.S. All authors have read and agreed to the published version of the manuscript.

**Funding:** This research received no external funding.

**Data Availability Statement:** No new data were created or analyzed in this study. Data sharing is not applicable to this article.

**Conflicts of Interest:** The authors declare no conflict of interest.

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
