# Peer review of "A Taxonomy of Product–Service System Perturbations through a Systematic Literature Review"

_jrfm, doi:10.3390/jrfm15100443_

Round 1

Reviewer 1 Report

In the introduction, the authors should give a detailed definition of the concept of  perturbation . Perturbation may refer more to the failure modes or barriers of the product service system. Why they use a new term to define PSS failure mode. The author needs to give a detailed argument and explanation. 

The author should further explain the research motivation of classifying the perturbation of product service system. 

Although the authors gave the definition of perturbation in the section 2.1, this definition is not clear. 

How they came up with figure one is not clear. 

In Table 2, the author ignores a lot of literature on failure modes of product service systems. This will affect the accuracy of their analysis results. 

How did the classification in Table 3 come about? What is the classification basis in Table 3? 

The author should add a new section to illustrate the theoretical significance of the study. 

In addition, they should conduct in-depth analysis of future research directions. 

Reviewer 2 Report

The correct definition of various phenomena and processes and the creation of their taxonomy is very useful for the development of science and for those who study a specific professional and scientific issue. This ensures the transfer of knowledge from the scientific to the practical field.

In the article, based on the study of 89 articles from the Scopus database, the authors proposed a taxonomy of product-service system perturbations. The authors have written an article that is useful for all researchers and article authors, teachers, experts who need to orient themselves in the issue of product-service system perturbations.

It is my personal experience that many reviewers of articles have serious reservations about publishing such articles. The reason is that super-advanced statistical methods are not used in articles of this type. I consider this article to be very useful for anyone who needs to orient themselves in the given issue, while the article also contributes to the development of scientific knowledge in this issue. The authors needed to be able to study a lot of professional and scientific literature and at the same time use a high level of analytical and synthetic thinking.

For this reason, I have no major comments and recommend the article for publication.

Reviewer 3 Report

Which is the need for arguing about perturbations in PSS? What about perturbations in both products and services contexts?

Is there any research raising the need to investigate such a domain? The need for this is not fully clear.

Are perturbations the only issue in providing PSSs? What about other risks and problems? For example the service paradox:

Brax, S., 2005. A manufacturer becoming service provider – challenges and a paradox. Manag. Serv. Qual. An Int. J. 15, 142–155. https://doi.org/10.1108/09604520510585334

Gebauer, H., Fleisch, E., Friedli, T., 2005. Overcoming the Service Paradox in Manufacturing Companies. Eur. Manag. J. 23, 14–26. https://doi.org/10.1016/j.emj.2004.12.006

Which is the relationship between perturbations and service paradox (or other issues)?

PSS design deserve a deeper analysis. Some valuable recent contributions in this domain (considering both the customer needs and the technical constraints) are:

Akasaka, F., Nemoto, Y., Kimita, K., Shimomura, Y., 2012. Development of a knowledge-based design support system for Product-Service Systems. Comput. Ind. 63, 309–318. https://doi.org/10.1016/j.compind.2012.02.009

Kimita, K., Mcaloone, T.C., Ogata, K., Pigosso, D.C.A., 2022. Servitization maturity model: developing distinctive capabilities for successful servitization in manufacturing companies Servitization maturity model 61. J. Manuf. Technol. Manag. 33, 1741–1779. https://doi.org/10.1108/JMTM-07-2021-0248

Kimita, K., Sugino, R., Rossi, M., Shimomura, Y., 2016. Framework for Analyzing Customer Involvement in Product-service Systems. Procedia CIRP 47, 54–59. https://doi.org/10.1016/j.procir.2016.03.232

Muto, K., Kimita, K., Shimomura, Y., 2015. A guideline for product-service-systems design process. Procedia CIRP 30, 60–65. https://doi.org/10.1016/j.procir.2015.02.188

"The term of barrier is popular in the field of PSS ". Please, provide some references about barriers in PSS domain.

Which kind of perturbation have already been investigated in literature? Dirsupted markets for example have been investigated recently.

Section 3

Sub-section 3.1 is useless. At least the title could be deleted.

Section 3.2

Why among the keywords terms as deviation, disruption, disaster have not been considered?

Through search queries 89 documents were obtained. 2 more were detected through other sources. However, 91 were the total amount after redundancies. This is not clear (authors didn't find redundances?).

At the end, 9 papers were selected. Why papers as the following were discarded?

Baines, T., Lightfoot, H., Benedettini, O., Kay, J.M., 2009. The servitization of manufacturing: A review of literature and reflection on future challenges. J. Manuf. Technol. Manag. 20, 547–567.

Pacheco, D.A. de J., ten Caten, C.S., Jung, C.F., Sassanelli, C., Terzi, S., 2019. Overcoming barriers towards Sustainable Product-Service Systems in Small and Medium-sized enterprises. J. Clean. Prod. 222, 903–921. https://doi.org/10.1016/J.JCLEPRO.2019.01.152

Table 3 is interesting. However, it seems to me that it does not propose something completely new. These are very similar to barriers identified in literature for PSS adoption. Which is the novelty of this study and of its results?

There are multiple contributions analysing the issues happening along the servitization path. It could be worth to analyse them:

Baines, T., Ziaee Bigdeli, A., Sousa, R., Schroeder, A., 2020. Framing the servitization transformation process: A model to understand and facilitate the servitization journey. Int. J. Prod. Econ. 221. https://doi.org/10.1016/J.IJPE.2019.07.036

Coreynen, W., Matthyssens, P., Van Bockhaven, W., 2017. Boosting servitization through digitization: Pathways and dynamic resource configurations for manufacturers. Ind. Mark. Manag. 60. https://doi.org/10.1016/j.indmarman.2016.04.012

Hsuan, J., Jovanovic, M., Clemente, D.H., 2021. Exploring digital servitization trajectories within product-service-software space. Int. J. Oper. Prod. Manag. 41, 598–621. https://doi.org/10.1108/IJOPM-08-2020-0525

Martinez, V., Neely, A., Velu, C., Leinster-Evans, S., Bisessar, D., 2017. Exploring the journey to services. Int. J. Prod. Econ. 192, 66–80. https://doi.org/10.1016/j.ijpe.2016.12.030

Oliva, R., Kallenberg, R., 2003. Managing the transition from products to services. Int. J. Serv. Ind. Manag. 14, 160–172. https://doi.org/10.1108/09564230310474138

Figure 3 is redundant after table 3.

In section 4, reference not reported in table 2 are used. This is strange since only the contributions selected in the lit rev analysis should be exploited to present the results.

Section 5 and 6 are well written. What could be the future research coming from this study. This is lacking. Usually literature analysis provide a list of future steps to be faced by researchers based on the results obtained. 

Reviewer 4 Report

Dear authors, thank you very much for your interesting paper. Despite the fact, that your findings can significantly broaden the theoretical background of the analysed field, there are some points which need to be addressed:

1. please, highlight the originality of the paper in the Introduction section

2. please, use up-to-date literature, there are only a few sources from the last 2-3 years. Please, add more references from this period to highlight the importance and topicality of the issue. 

3. section 3 is too structured (3.1 is really not necessary). Please, revise and make the paper more consistent.

4. conclusions are too weak - please, highlight your crucial findings & value added to the paper; add limitations of your study and future research challenges. 

Round 2

Reviewer 1 Report

The authors have addressed my previous concerns. 

Author Response

Thank you so much for the acceptance! We appreciate a lot for your help and support! 

Reviewer 3 Report

The paper has been improved.

Literature about PSS design should be still improved. There are multiple contributions in the recent literature that could support your research in this domain. A review is provided in the following reference:

C. Sassanelli, G. Pezzotta, F. Pirola, M. Rossi, and S. Terzi, “The PSS Design GuRu Methodology: Guidelines and Rules generation to enhance Product Service Systems (PSS) detailed design,” J. Des. Res., vol. 17, no. 2/3/4, pp. 125–162, 2019, doi: 10.1504/JDR.2019.105756.

In addition, there is an entire set of related contributions enriching this domain (concerning the grounding approach, the integration of information systems to manage product and services, the knowledge management, etc.). 

Author Response

Thank you so much for your precious comments. We have revised the paper based on your comments. Please see the attachment.

Reviewer 4 Report

Dear authors, the paper was revised following my comments and recommendations, which were fully accepted. 

Author Response

Thank you so much for your acceptance! We appreciate a lot for your precious suggestion.